# Fertility Deterioration in a Remediated Petroleum-Contaminated Soil

**DOI:** 10.3390/ijerph17020382

**Published:** 2020-01-07

**Authors:** Verónica Isidra Domínguez-Rodríguez, Randy H. Adams, Mariloli Vargas-Almeida, Joel Zavala-Cruz, Enrique Romero-Frasca

**Affiliations:** 1División Académica de Ciencia Biológicas, Universidad Juárez Autónoma de Tabasco, Carretera Villahermosa-Cárdenas Km. 0.5 S/N, Entronque a Bosques de Saloya, CP 86150 Villahermosa, Tabasco, Mexico; tazvro@hotmail.com (V.I.D.-R.); mary_gstark@hotmail.com (M.V.-A.); enriqueromero7@gmail.com (E.R.-F.); 2Colegio de Posgraduados, Periférico S/N, 86500 Heroica Cárdenas, Tabasco, Mexico; joel_zavala@yahoo.com.mx

**Keywords:** compaction, physical-chemical treatment, residual hydrocarbons, soil profile, toxicity, water repellency

## Abstract

A soil that had been remediated by soil washing and chemical oxidation was evaluated, comparing it to an uncontaminated control soil ~30 m away. Profile descriptions were made of both soils over a 0–1 m depth, and samples were analyzed from each soil horizon. Samples were also analyzed from surface soil (0–30 cm). The control soil (a Fluvisol), had several unaltered A and C horizons, but the remediated soil presented only two poorly differentiated horizons, without structure and much lower in organic matter (<0.5%). In surface samples (0–30 cm), the bulk density, sand-silt-clay contents, field capacity, organic matter, and porosity were different with respect to the control (*p* > 0.05), and there was much greater compaction (3.04 vs. 1.10 MPa). However, the hydrocarbon concentration in the remediated soil was low (969.12 mg kg^−1^, average), and was not correlated to soil fertility parameters, such as porosity, organic matter, pH, moisture, field capacity or texture (R^2^ < 0.69), indicating that the impacts (such as compaction, lower field capacity and moisture content) were not due to residual hydrocarbons. Likewise, acute toxicity (Microtox) was not found, nor water repellency (penetration time < 5 s). It was concluded that the fertility deterioration in this soil was caused principally from the mixture of upper (loam) and lower (silty clay to silty clay loam) horizons during remediation treatment. Another important factor was the reduction in organic material, probably caused by the chemical oxidation treatment.

## 1. Introduction

Pollution in southeastern Mexico is attributed to petroleum activities because of hydrocarbon spills from retention dams or pits, oily water filtration through the dam containment dikes and/or foundations and by broken oil pipelines. The latter is an increasing problem due to the corrosion of pipelines and the dispersion of spilled oil by surface runoff [1].

In Mexico, as in most countries, environmental regulations and norms are developed to remediate oil-contaminated sites. These regulations are focused on reducing the hydrocarbon concentrations in soil. This is based on the supposition that the main problems in such sites are toxicity caused by the oil. By reducing the contaminant concentration to some legal limit, this effect will be eliminated or sufficiently mitigated. In response to such legislation, remediation technologies are mainly focused on reducing the hydrocarbon concentration in soil. However, the consequences of oil spills in rural areas are mainly changes is soil physical and chemical properties, in such a way that fertility or primary productivity is significantly impacted [2,3,4,5]. None-the-less, environmental agencies do not take soil fertility into account, and regulations and norms generally lack specifications to restore fertility in oil-contaminated soils [6]. This is one of the main uncertainty factors of remediation effectiveness for oil-contaminated sites in a rural environment. Thus, restoring soil fertility (and its ability to grow pasture, crops and natural flora) is not guaranteed by only reducing the hydrocarbon concentration to legally permissible limits.

Among the different remediation techniques used to treat oil-contaminated sites, there are many thermal, physical-chemical and biological remediation methods [7]. Most of these methods are only limited to achieving permissible levels of hydrocarbons as established by laws and regulations. They do not take into account that the main objective of soil remediation/restoration: recovering its fertility and avoiding greater environmental damage [6]. Several studies have evaluated the effectiveness of these remediation techniques, and in many cases the results are not positive. Even though some achieve the maximum permissible levels established by law, the physical-chemical properties are still compromised and sometimes the residual hydrocarbon concentrations still surpass the permissible levels [2,3,8].

Many researchers consider that bioremediation methods are superior to physical-chemical methods for cleanup of hydrocarbon-contaminated soil (see for example, [9,10]). Although physical-chemical treatments are widely used due to the short time required for the reduction of high hydrocarbons concentrations, many researchers have considered that the effects on soil fertility are very negative [11,12,13]. This is usually believed to be associated with the use of oxidizing agents, soil stabilizers and surfactants. Soil stabilizers may partially sterilize the soil due to changes in soil pH. Surfactants can break apart cell membranes and kill important microbes and other biota. Also, oxidizing agents, such as hydrogen peroxide, may actually consume organic material other than the contaminants, including natural soil organic material and cells. None-the-less, there have been very few studies that consider how general remediation practices may affect soil fertility. Thus, these cleanup methods may be inadequate to retain or actually restore soil fertility in contaminated sites, as a possible cause of these negative impacts. Most bioremediation projects do not suffer from these kinds of negative processes. However, any technology aimed at oxidizing an organic contaminant will not only oxidize the organic contaminant, but also normal soil organic material, and may damage the soil. This is true whether using mineral catalyzers in combination with chemical oxidants, or biological catalyzers—enzymes—in bioremediation.

In the present study, the effects and probable causes of fertility deterioration in an Orthofluvic Fluvisol that was contaminated with crude oil and remediated with an ex situ physical-chemical treatment were identified, using a weighted averages approach and considering the entire depth of the excavated and treated soil. Alternative and additional field scale treatment techniques are proposed so that soil fertility is conserved during the remediation procedures.

## 2. Materials and Methods

### 2.1. Control and Remediated Soil Collection

Control and remediated soil samples were collected at the same site in Rancheria Los Cedros, Cunduacán Municipality, Tabasco, in southeastern Mexico (18°03′20.9″ N and 93°03′20.8″ W). The land use at this site was predominantly pasture for cattle raising. This site has a humid, tropical monsoon climate (average temp. ~27 °C, annual precipitation ~2000 mm).

On June 26th, 2006, the site was contaminated by a crude oil leak from a 24 diameter oil pipeline. Soil restoration was not initiated until 2007 by a private contractor hired by the state-run oil company, Petróleos Mexicanos. The remediation company was not willing to divulge information on the method used, claiming that it was proprietary information (an industrial secret). However, the property owner and one company employee (unofficially) stated that they used an ex situ physical-chemical treatment consisting of soil washing with surfactant and chemical oxidation using hydrogen peroxide [11]. For remediation, the contaminated soil was excavated to a depth of three meters and transferred to a treatment cell nearby. After treatment to reduce the hydrocarbon concentration to environmental norm, the soil was returned to the excavation. No post-treatment was made at this site (such as tilling, addition of organic soil conditioners, etc.). There was a five year time lapse between remediation and site sampling.

Samples were collected in the remediated area and control samples consisted of uncontaminated soil of the same type as that which was contaminated and remediated, approx. 30 m away. Control (0–118 cm) and remediated (0–108 cm) soil profile descriptions were performed, one in each area. The soil profiles were made according to [14] and included color in the field using the Munsell color code for soils, apparent texture, structure, consistency, presence and type of pores, estimated permeability, mineral agglomerations, presence and size of roots, presence of mesofauna (ants, spiders, beetles, worms), presence of stones, gravel and anthropomorphic materials (such as plastics). For each soil horizon (Ho) found in the soil profile, a 200 g sample was collected. Additionally, five surface samples (0–30 cm), about 20 m apart, were also collected from each area. All samples were dried, ground and sieved at a laboratory facility. Soil texture was classified according to procedures of the United States Department of Agriculture [15] and the Food and Agriculture Organization of the United Nations [16].

Photographs of the soil profiles were taken in the field in mid-December, under a partially cloudy sky about midday, at approx. 30 °C and 80% relative humidity. The photographs were taken with a Lumix DC Vario digital camera (Panasonic AVC Networks Xiamen Co. Ltd., Xiamen, China) with a 10 megapixel capacity. Images were imported into MS Word documents and slightly modified using the MS Office software. The modifications consisted of slight changes in brightness and contrast, as well as saturation, to improve clarity and more closely resemble what was actually observed by eye in the field.

### 2.2. Soil Physical-Chemical Analysis

The following physical-chemical parameters were investigated for both, surface soil and soil horizon samples: pH was determined using a Orion 3-Star Plus pH Benchtop Meter potentiometer (Thermo Scientific^®^, Waltham, MA, USA). Organic matter (OM) was analyzed by applying the Walkley and Black [17] soil organic carbon oxidation method using potassium dichromate, and Texture (T) was determined by the Bouyoucos hydrometer test after applying hydrogen peroxide, according to [18]. Bulk density (BD) was determined according to the graduated cylinder method of Domínguez and Aguilera [19], and particle (solid) density (SD) values were measured as per the NOM-021-SEMARNAT-2000 method [18]. For moisture content (soil Humidity, H) and porosity, a MB35 Basic Moisture Analyzer (OHAUS, Parsipanny, NJ, USA) and a mathematical relation between BD/SD were used, respectively.

### 2.3. Field Capacity

Based on the Colman column principle [20,21], field capacity (FC) was measured. For each sample, 100 g of soil were placed into a bottom-perforated plastic cup and placed in an open top water container for two hours. Afterwards, the wet samples were drained in a steel mesh for 24 h. Subsequently, the samples were weighed and oven-dried at 60 °C for 24 h. Field capacity was calculated by gravimetric analysis.

### 2.4. Hydrocarbon Concentration

The Total Petroleum Hydrocarbon (TPH) concentration in soil was determined by US EPA method 418.1 according to [2] and according to the Environmental Protection Agency [22]. Two grams of soil sample and 0.5 g of anhydrous sodium sulfate were mixed with 25 mL of perchloroethylene as an extraction solvent. After 48 h of extraction, the mixture was filtered using Whatman #40 Filter Paper. Finally, the filtered mixture absorbance was measured using an Infracal TOG/TPH Analyzer (Wilks Enterprise, Norwalk, CT, USA) and the TPH concentration was calculated using an extra-heavy petroleum calibration curve [23].

### 2.5. Acute Toxicity

Acute toxicity was measured by the Microtox bioassay using marine bioluminescent bacterium *Vibrio fischeri* as a test organism, based on [24,25]. This bioassay consists of bioluminescent light emission inhibition in the bacterium under toxic stress. Using the resulting readings, EC_50_ values were determined. EC_50_ is the Effective Concentration 50: the concentration of sample which reduces the bioluminescence by 50%. These EC_50_ values were then calculated as Toxicity Units (TU). The following relation was used: TU = (1/EC_50_), with the EC_50_ value stated as a proportion. For example, if EC_50_ = 10,000 mg/L = 1%, or as a proportion = 0.01, TU= 1/EC_50_ = 1/0.01 = 100.

### 2.6. Water Repellency Analyses

The Watson and Letey [26] Water Drop Penetration Test (WDPT), as modified by [2], was used to determine water repellence persistence in soil and classified as per Dekker and Jungerius [27]. Dried soil was added and uniformly flattened into an empty Petri plate. Afterwards, five small droplets of water were placed on the soil surface and the time for absorption into the soil was measured.

### 2.7. Soil Compaction Test

Soil field compaction was measured using a 30° circular stainless steel penetrometer (Dickey-John, Shoreview, MN, USA) equipped with a manometer and a guide rod according to Duiker [28].

### 2.8. Data Analysis

Previous to our study, the site had been remediated, excavating contaminated soil to a depth of 3 m. Thus, it was necessary to compare the two soils (remediated and control) over a 0–3 m depth. However, in the field, the remediated soil was so compacted that hand digging to that depth was not possible. Accordingly, the soils were compared using the following considerations:

In the 1950s artificial levies, about 4 m high, were built on both sides of the nearby Samaria River. Considering that these hydraulic structures reduced the site flooding processes, it is probable that no apparent changes with depth have occurred in the control soil since that time. Hence, the deepest final four soil horizons (5–8) were selected as the most representative and used to obtain the 1–3 m depth values for the control soil using the equation below:$$WA=\frac{\Sigma \left(\left(Ph*z\right)*Ph\right)}{\Sigma \left(Ph*z\right)}$$
where *WA* = weighted average (units depend on the parameter analyzed), *Ph* = value of the parameter for a certain horizon, *z* = horizon depth.

To make comparisons of weighted averages between the remediated and control soil, the values for depths of 1–3 m were estimated to be based on the value in the lowest soil horizon in the remediated soil. Thus, weighted averages were calculated using data from the 0–1 m depth, and adding the 1–3 m depth based on data from the final horizon in each soil profile.

For remediated soil, the 1–3 depth values were considered to be the same as the deepest soil horizon (Ho-2), due to its similarity with other field horizons, and to probable unchanging conditions on deeper soil layers since the last remediation project.

Additionally, a statistical variance of means analysis was performed to determine statistically significant differences between control and remediated surface soil samples. To do so, a Student’s *t*-test (using Excel 2007) considering a 95% confidence interval was performed. Additionally, possible correlations between residual TPH in the soil and various soil properties were evaluated. For this the data from five samples of surface soil was used. Each parameter was evaluated independently vs. TPH. Individual graphs were made of TPH vs. each of the other parameters evaluated using the Microsoft Office Excel software and linear regressions were carried out using this same software. There was no data where alternative correlations (exponential, potential, logarithmic) gave better correlations.

## 3. Results

### 3.1. Control Soil Profile

A description of the control soil profile is shown in Figure 1. Classified as an Orthofluvic Fluvisol (Geoabruptic) (FLof (go)) according to the International Soil Classification System [16], the soil appeared to be stratified with fluvial material and had >0.2% of organic carbon (OC) in deeper layers (>71 cm depth). This quantity of OC is greater than 25% with respect to the OC located at the mid soil layers (25 to 71 cm depth). Furthermore, stratification, texture variation and light-to-dark layer alternation were also observed in the soil profile. From horizons 5 to 8 (50.5–118 cm) a buried soil (Cb horizons), with similar characteristics to the first four superficial layers (Ap and C horizons), was found with varying sand and clay content as well as different organic matter concentrations. Soil genesis was attributed to hydrological changes and flooding/sedimentation processes prior to 1955. The site, located inside the Mezcalapa River delta and its tributary, the Samaria River, was so vulnerable to flooding [29], that authorities were forced to build river flooding defenses (dikes) along the Samaria River in 1956. Since then, the site has been generally free from fluvial flooding and alluvial accumulation over the years.

Control soil physical and chemical analyses results are shown in Table 1. While the soil organic matter concentration was classified as medium (>1.6%) in horizon 1, a low to very low content of organic matter (<1.6%) was determined in the other horizons (2 to 8). Silt and clay fractions were predominant (>50%) in five of the eight horizons studied, and especially so in three of the four deepest horizons. Texture was classified as loam, silty clay and silty clay loam. Lower bulk density values were obtained in the surface horizon (typical of uncompacted soil), as well in deep horizons 5, 6 and 8, characteristic of clayey soils. The pH was mildly alkaline (at or near pH = 8) in all horizons except the surface horizon, which had a neutral pH. The field capacity had a range of 28–36%H and the water repellency null (WDPT 0.48–3.26 s vs. 5 s), in all horizons.

### 3.2. Remediated Soil Profile

Remediated soil was classified as a Spolic Technosol (Hyperartefactic) formed from solid substances substantially modified by human activity, which occupied more than 50% of soil total volume and had a thickness of over 20 cm. Only two horizons were observed as seen in Figure 2, where the first one spans a depth of 63 cm, similar to what was previously described by [11] at a depth of 0–47 cm. Field observations showed that both horizons were made of very hardened, cracked and agglomerated anthropogenic material along with gravel fractions and small, whitish, lumpish and hard carbonate-like incrustations. Moreover, the soil was heavily compacted (grass roots only penetrated to depths of 3–10 cm) and slight hydrocarbon odors (from a depth of 63 cm downwards) were perceived in the site. Small, high-density plastic membrane pieces were also found in both horizons. Considering the aforementioned materials, it was concluded that significant soil profile changes took place during the remediation process and led to the formation of a different soil group in comparison to the control site.

Remediated soil horizons did not show any relevant differences in physical and chemical characteristics, (Table 2), where both had low concentrations of organic matter (<0.5%) and abundant clayey and silty fractions (being classified as a clay loam). Mildly alkaline properties were defined by the pH (7.4–8.5). Almost identical field capacity and particle density values were determined for both horizons. Although null water repellency classification (WDPT < 5 s) was found for both horizons, water repellency was slightly more in horizon 2 (1.97 s).

### 3.3. Control Soil vs. Remediated Soil Profile

#### 3.3.1. Physical-Chemical Properties

Weighted averages between 0–3 m were employed for this analysis. Soil texture was practically identical for both soil profiles (Figure 3a), indicating that there were no important changes in texture when the entire depth of remediated soil is considered and compared to the control (even though in surface soil a large change was noticed, Figure 4a). However, comparing values in Figure 3b there were slight differences. Bulk density was about 1/10th greater in the remediated soil than in the control soil and the particle density slightly less, resulting in a porosity of about 1/10th less. Likewise, field capacity was about 1/10th less but the moisture content was much less, being only a little more than one-half of that found in the control soil (Figure 3b), similar to that found in the surface soil (Figure 4b).

When comparing the entire soil profile, the remediated soil also has about two-thirds less organic material (Figure 5). This is probably due to the use of hydrogen peroxide treatment during the remediation process, which not only oxidized the hydrocarbons, but a large part of the natural organic material in the soil. It is worth mentioning that this may not be a phenomenon limited to chemical oxidation treatment only. The use of bioremediation, supplemented with inorganic nutrients (especially nitrogen) may stimulate the soil bacteria to consume and mineralize natural organic substances in the soil as well as petroleum hydrocarbons.

The reduction in soil organic matter (OM), as well as mixing of finer subsurface horizons into the surface horizons, may be at least partially responsible for the increased compaction observed in the remediated soil (see Figure 6), and correspondingly, the reduced field capacity and moisture content, and finally the reduced depth of root penetration. Organic matter decrease has also been shown to be related to irregular plant growth [30,31].

#### 3.3.2. pH and Water Repellency

Soil pH was determined as mildly alkaline (7.4–8.5), which does not affect soil fertility. WDPT values were 1.88 and 1.81 s for control and remediated soil, respectively, being both well below null soil water repellency limits (WDPT < 5 s).

### 3.4. Control vs. Remediated Surface Soil

No texture variation was observed when comparing soil profiles (0–3 m), however, there was much greater compaction and poor root penetration in the remediated soil (Figure 6). To study this in greater detail, surface soil (0–30 cm) was evaluated due to its agricultural importance, and because most vital conditions for crop growth are found in this depth [15]. Each condition is described in the following subsections.

#### 3.4.1. Physical-Chemical Properties

In contrast to the 0–3 m soil profiles, the surface soil samples showed large differences in texture between the control and remediated areas. The amount of sand was more than one-third less, while the finer fractions were more than a third greater (Figure 4a). This was probably due to the mixture of upper with lower soil horizons during the remediation project (excavation up to three meters and mixing during treatment). The bulk density and particle density were about 1/10th greater in the remediated soil than in the control soil, and likewise, the porosity was almost 1/10th less. The field capacity was a little more than 1/10th less, but the moisture content was severely impacted, being almost one-half less than in the control soil. While field capacity, organic matter, porosity, moisture and texture for both surface soils (0–30 cm) were significantly different (*p* < 0.05), bulk density was not (*p* > 0.05).

As seen in Figure 5, when comparing superficial soil (0–30 cm), the remediated soil had about three times less organic matter. This is also noted when comparing the first two soil horizons in the control soil (Ho-1: 0–15 cm, Ho-2: 15–25 cm) with the first horizon in the remediated soil (a well-mixed horizon of 0–63 cm) which had about four times less organic material. Scarce OM content (<0.5%, more than two-thirds less than control surface soil) was observed also, and was considered to a factor likely responsible for reduced moisture (about one-half of that in the control soil), and field capacity (between 1/10th and 1/5th that found in control soil) (Figure 4 and Figure 5). It is likely that low OM content may have also contributed to soil compaction (Figure 6), increased bulk density and lower porosity (Figure 4). These kinds of factors have been shown to be vital for vegetative growth [31]. Biological treatments (like soil conditioners) were not considered for post-remediation restoration at the site, but could have contributed to soil organic matter enrichment, increased moisture content and possibly avoided the severity of these conditions, as found by other authors [23,30,32,33,34].

#### 3.4.2. pH and Water Repellency

Although soil samples pH difference was not statistically significant (*p* > 0.05), the remediated soil appeared to be mildly alkaline (7.4–8.5). In the remediated soil, a slight pH increase could be attributed to the alkaline chemical products used to treat spilled petroleum by products such as residual oily substances and/or salty process water. However, these pH values are not considered to be a soil fertility problem for most agricultural uses [18].

Likewise, the water repellency difference was not deemed as statistically significant (*p* > 0.05), but a slight tendency to increase absorption time in remediated soil was observed. The latter could be due to residual hydrocarbons in soil that causes slight soil water repellency but does not modify its properties very much. None-the-less, all absorption time values were well below null soil water repellency classification (WDPT < 5 s).

#### 3.4.3. Acute Toxicity

No test organisms’ bioluminescent light emission inhibition pattern using greater sample concentrations was observed, either in control soil or remediated soil. Thus, both soils were classified as nontoxic, at least in the 0–30 cm depth. For some samples, bacteria bioluminescent light emission was even slightly stimulated.

### 3.5. Soil Penetration Resistance (Compaction)

Penetration resistance was measured in situ (Figure 6), in the month of June, considering Duiker’s [28] suggestion that sampled soil should be at maximum soil field capacity (achievable after a 24 h rainwater immersion). This is a standardized method of Pennsylvania State University/US Dept. of Agriculture. Samples were taken seven days after a week-long rainfall. As seen, there was more variability in the remediated soil, probably due to the uneven manipulation during excavation, treatment, and refilling of the excavated area. However, even with this variability the trend becomes clear, especially at depths greater than 20 cm.

The control soil was not considered to be compacted, in fact it was almost optimal for crop growth, 1–3 MPa, according to some authors [15,35]. However, the remediated soil showed an increasing tendency for compaction as depth increased, resulting in a 3.04 MPa soil compaction in the first 30 cm as seen on Figure 6. This last value indicates that root growth, as well as nutrients and water absorption conditions would be disadvantageous for proper crop growth [33]. A previous study [11] reported compaction up to 7.2 MPa in surface soil.

### 3.6. Residual TPH Concentrations and Soil Physical-Chemical Properties Relation

The data presented in this section deal with surface soil samples (0–30 cm), using five replicates for each area (control, remediated soil). In this study, the control and remediated soils were considered to be different statistical populations due to the very different soil profiles observed in the field, the different histories at the site (contaminated and then remediated vs. never contaminated) and considering the large differences observed in texture in the surface soil (0–30 cm) samples.

The difference in residual TPH concentrations between the soils studied were calculated as statistically significant (*p* > 0.05), being greater in the remediated soil (maximum of 1143 mg kg^−1^) than that of the control soil (maximum of 591 mg kg^−1^, arguably of vegetative origin). The average of TPH values in the control and remediated soils were 181.70 and 969.12 mg kg^−1^ respectively. Thus, it appears that this difference is due to a low concentration of residual petroleum hydrocarbons in the remediated soil left after the remediation process. None-the-less, these TPH concentrations were much less than the maximum permissible level (3000 mg kg^−1^) established by the Mexican environmental norm for agricultural soils [6].

In many sites, residual TPH concentrations can be damaging in soil, impacting physical-chemical properties and resulting in a reduced soil fertility [3,5]. However, this does not appear to be the case at this site, as shown by the general absence of correlations between the TPH concentrations vs. soil physical-chemical properties for fertility (Table 3). In general, there was only a very poor correlation between TPH and most soil properties in the remediated soil (R^2^ = 0.0001–0.5329); only the WDPT values were somewhat correlated (R^2^ = 0.6889). Even so, the actual WDPT value was still lower (1.08 s) than that for “slightly” water repellent classification (WDPT < 5 s), and the soil was classified as non-repellent according to the scale proposed by Dekker and Jungerius [27].

In fact, the control soil showed a higher correlation between hydrocarbon concentration and some fertility parameters, with R^2^ = 0.6400, 0.7500 and 0.8100, for porosity, particle density and bulk density, respectively. This was probably not really due to residual hydrocarbons, but hydrocarbons of vegetable origin instead, reducing soil density and improving soil structure.

## 4. Discussion

### 4.1. Probable Causes of Soil Fertility Deterioration at This Site

A summary comparison of fertility parameters between the control soil and remediated soil is presented in Table 4. As shown, in the remediated soil, the amount of course grain material (sand) in the surface soil is only about 3/5th he amount in the control soil. In these fine textured, alluvial soils, this may be a cause of compaction, poor water infiltration, and possibly, poor root penetration [3,4,30,35]. The increase in fine grain material with a high CEC could be favored by this change in texture. However, in these kinds of soils, the CEC is already high and usually not much of a concern (200–450 meq kg^−1^) [3,4]. Therefore, is unlikely that the ability of the soil to retain and provide nutrients to plants was impacted considerably. Similarly, the amount of organic material in the remediated soil (0–30 cm depth) was three times less. This could also result in problems with soil compaction, as well as field capacity [3,30,33].

Indeed, in the following lines in the table it can be seen that field capacity, porosity, bulk density and compaction were adversely affected in the remediated soil. Especially damaging was compaction, being two to three times greater in the remediated soil, and above 3 MPa at 30 cm depth. Soil moisture was also very adversely affected, being roughly one-half that of the control soil. Although the soil-water-plant relationship is frequently impacted in petroleum-contaminated soils [2,3,5,8], in the remediated soil no water repellency was found (WDPT < 2 s, classified as null as per [27]). Thus, it would appear that the low soil moisture content is most likely due to the compaction caused by the textural changes resulting from mixing the clayey subsoil with loamy surface soil during the ex situ remediation process. Additionally, the reduction in soil OM (which was 15% less in the remediated soil), may have reduced the field capacity somewhat.

It needs to be mentioned that in many remediation projects the use of heavy machinery may also alter soil bulk density. Typically, when soil is excavated, the bulk density decreases due an increase in soil volume, usually on the order of about one-third. This is because the excavated soil is no longer compressed by the lateral forces that were present in the original, unexcavated condition. When left alone, the soil tends to settle, and over a long time period, may return to a bulk density near that of the initial, unexcavated soil. But in some cases, the equipment operators will purposely compact the soil with heavy machinery to assure that all of the treated soil fits in the excavation. At sites with industrial or commercial use, this may be beneficial for construction purposes. However, when the land use is for farming or pasture it is usually detrimental, reducing water infiltration, root penetration, and proper gas exchange in the soil.

At this site compaction due principally to the use of heavy machinery is unlikely. Previously, another research group performed in situ bioassays with radish plants at part of this site, comparing the remediated area with the control area [12]). Prior to planting, the soil was fertilized and tilled manually. No adverse effects in germination, establishment, or plant vigor was observed, but the size of the radish bulbs were drastically reduced in the remediated soil. It is likely that this was due to the fine texture of the surface soil and natural settling post-tilling. Thus, even if the compacted soil is physically tilled to overcome compaction, it appears to settle back into a denser state than the control soil.

In the last two lines of the first section of Table 4, one can observe how the adverse soil properties of the remediated soil affected the soil biota. In the control soil, roots were found at all depths of the soil profile (up to 118 cm deep), but in the remediated soil, roots were only found at a depth of 3–10 cm. Likewise, in the control soil, insects were found throughout, but in the remediated soil, only in the first soil horizon (0–63 cm) and mostly at the very top of this horizon (0–15 cm).

In the following section of Table 4, those factors which were probably not related to soil degradation are presented. Firstly, it is notable that there was no evidence that the residual hydrocarbon concentration in the remediated soil was related to any of the parameters measured (R^2^ < 0.7). The final concentration was relatively low (<0.1%) and more than three times below the national environmental norm [6]. It is unlikely that this factor could be the cause of reduced biota in this (remediated) soil. As previously mentioned, the water repellency was null and no toxicity was observed in either control soil or remediated soil. Likewise, the pH was only slightly higher in the remediated soil, and in the same range as the subsoil in the control soil. It was considered neutral—mildly alkaline according to the national norm for soils [18], and therefore unlikely to be a cause of soil deterioration in the remediated soil.

In Table 5, additional factors to consider in future studies are presented. These are factors which were not measured directly in the present study, although some indications of their importance can be gained from the parameters that were measured, or by reference to other studies. Some authors have mentioned that the ability of the soil to maintain and supply nutrients to soil plants may be affected in contaminated soils [3,8]. This is usually measured as the cation exchange capacity. In the current study, it is improbable that this factor was affected in the remediated soil. Reduced CEC and field capacity, as well as the development of water repellency are usually caused by the formation of thin laminates of residual, degraded oil on soil aggregates [2,3]. However, in the present study, no water repellency was found. Therefore, problems with CEC are not expected. Additionally, in surface soil, the amount of clayey, CEC-rich minerals was increased 24%. None-the-less, in future studies in other soils, measuring CEC as well as field capacity and water repellency could be very useful (especially in sandy soils).

Salinity is also a factor that may be important in some contaminated/remediated sites. If oil is spilled before the process water has been removed in a dehydration plant, the spilled liquid will contain oil and also salty water; and the affected soil may present salt stress. In the present study this was not the case, and the high pH (often co-associated with process water) was not observed.

Other, more precise measurements of the biota at contaminated/remediated sites may also be warranted in future studies. In some sites, the adverse effects of high pH from soil stabilizers, from the biocidal properties of ionic surfactants, and strong oxidizing agents such as hydrogen peroxide, may kill many of the soil organisms. This may include microorganisms important to soil nutrient flows and availability, plant rhizomes, and soil mesofauna such as earthworms, nematodes, beetles and ants. At this site, however, it is unlikely that the soil would not have recovered post-remediation. This would be due to the generally high CEC of the alluvial soils in this region [3,5], the five year time lapse between remediation and site sampling, and the humid, tropical monsoon climate (average temp. ~27 °C, annual precipitation ~2000 mm), which are ideal conditions for recovery. None-the-less, it would be useful to measure such parameters as total heterotrophs, microbial respiration or other microbial enzyme activities [23,33]. Likewise, other bioassays with plants would be useful to clearly demonstrate recovery or lack thereof [23]. For example, at this site, another research group had previously run an in situ bioassay with radish plants. In that study, practically no effects were found on germination, establishment or plant vigor, but the bulb diameter in the remediated soil was significantly less, probably indicating problems from compaction [12].

Using data available in Table 1 and Table 2, the control and remediated soil horizons distribution was illustrated (Figure 7). While the first four control soil horizons showed a higher sand content (43–67%), three of the lower four soil horizons were much richer in clay and silt (only 7–15% sand), and the exception (Ho-7) was only 4 cm thick. The remediated soil profile showed that the two horizons had an intermediate sand content (20–30%). This is almost certainly due to the mixing of upper and lower soil horizons during the remediation project, which led to a texture modification in the remediated surface soil (Figure 4a). Thus, a clay loam texture (vs. a loam) was found at the surface, probably resulting in greater compaction and bulk density, lower soil porosity, and less water infiltration in the surface soil, where most of the root growth takes place.

Reduction of OM content was accounted as the second negative impact of soil remediation, probably due to the oxidizing chemical used. This may have contributed to soil compaction, reduction in soil moisture and water infiltration rate. Vegetative growth was very limited also, possibly, not only by these factors, but by the lack of nutrients that could be derived from slow decomposition of the soil organic matter.

Although the use of organic fertilizers as post-remediation alleviators to the previous problems was not considered in this remediation project, it is very likely that they would have ameliorated negative soil impacts, as found in other studies [23,30,32,33,34]. It is noteworthy that any remediation technique that depends on crude oil oxidation could result in vegetative OM-content oxidation as well. This could happen when either inorganic catalysts are used (for example, in the Fenton reaction used in most chemical oxidation remediation projects) or with biocatalysts (like the enzymes in the bacteria employed in bioremediation).

### 4.2. Recommendations to Avoid Soil Fertility Deterioration in Remediated Soils with Agricultural Land Use

Based on observations in the field, and considering good agricultural and remediation practices, these recommendations are made in order to avoid soil fertility deterioration and properly treat contaminated soil:(1)Excavate all contaminated soil from the first 30 cm. Soil horizons (mainly A or O) within these depths should be treated as a whole.(2)Excavate all contaminated soil ranging from 30 cm to where visually affected soil is still found. Soil horizons of this second excavation (mainly B or C) should be treated as a whole, but not mixed with the previous mixture.(3)Assuming the soil remediation is completed, the second excavated soil layer should be returned first to the remediation site and finally, on top of this, the treated surface soil (0–30 cm).

These three steps could be used to avoid mixing subsoil layers higher in clay and lower in organic material with the more fertile surface soil.

(4)Soil conditioners and a mid-term vegetative cover (2–3 years) should be incorporated at the remediation site. This setup can add organic matter and nutrients, improve soil structure and improve moisture retention capacity [23,30,33]. It may also replenish microbial populations and other soil biota if it was damaged by chemical treatments. Therefore, it is strongly recommended after physical-chemical or biological remediation [34,36].

These recommendations are made only for soils with agricultural or pasture use and specifically for soils with conditions similar to that found in this study (original surface soil with better texture and/or organic matter content than the subsoil).

### 4.3. Applicability of Recommendations to Other Sites

The results of this study may be applicable to many other sites with similar conditions. These would be sites that have been contaminated by petroleum spills in a rural environment where the land use was for farming or pasture. This is very common in many petroleum producing areas in tropical and temperate climates. Other conditions that need to be similar to apply the results of this study would be the kind of soil, especially with respect to differences in surface soil and subsoil. At this site the surface soil had a much higher organic matter content, and a coarser texture than the subsoil. This condition results in greater fertility in the surface soil than in the subsoil. This tends to be the rule for soils [15], especially with farming or pasture use, although there may be some exceptions.

Important exceptions would be some very weathered tropical soils with very poor fertility, such as many Ferralsols [16], which are prevalent in tropical South America, and central Africa. Another exception would be some soils from arid zones which have not had sufficient rainfall or time to form an A horizon with a large amount of organic material. Additionally, in some sandy soils, especially in coastal areas of very recent deposition in which there has been almost no accumulation of organic material in the A horizon these results may not apply. Likewise, this may be the case in some areas of very frequent alluvial deposits near large rivers, where there is very little soil horizon development between alluvial depositions. This may occur is some petroleum producing areas of the upper Brahmaputra River in Assam, India, in the Niger River Delta, and in the Orinoco Delta in Venezuela [23]. It is also worth mentioning, that there may be some sites where other human activities may have disrupted the usual pattern of soil horizons previous to a spill. This may be observed very locally where there has been some construction, such as right next to separation batteries, valve stations or on well pads. In all of these areas, it would not be important to avoid mixing surface and subsurface soil since they may not have much of a difference in terms of fertility (even prior to remediation). However, even in these soils, the addition of organic matter and nutrients would very likely help overcome OM loss caused by chemical or biological co-oxidation with petroleum hydrocarbons.

## 5. Conclusions

Remediated soil fertility damage was probably not due to residual hydrocarbons at this site, or negative effects of chemical remediation, but instead, to the mixing of lower subsurface soil horizons (silty clay to silty clay loam) with upper horizons (loam) during the remediation process. Also, to organic matter loss, probably due to the chemical oxidation technology used (where crude oil and circumstantially, soil organic matter, could both be oxidized). In the present study, better remediation techniques are suggested so that soil fertility does not deteriorate in surface soil during the restoration processes, and the site may be used for crops or pasture. Furthermore, post-remediation restoration techniques are suggested to increase organic matter and restore soil properties related to organic matter loss. These recommendations are important since the majority of remediation projects, even for rural areas, do not explicitly consider conservation and restoration of soil fertility.

## Figures and Tables

**Figure 1 ijerph-17-00382-f001:**
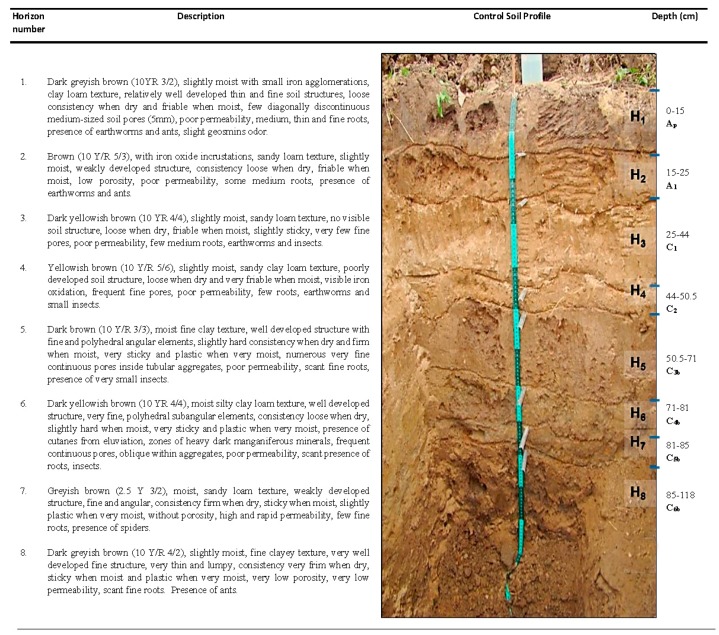
Control soil profile.

**Figure 2 ijerph-17-00382-f002:**
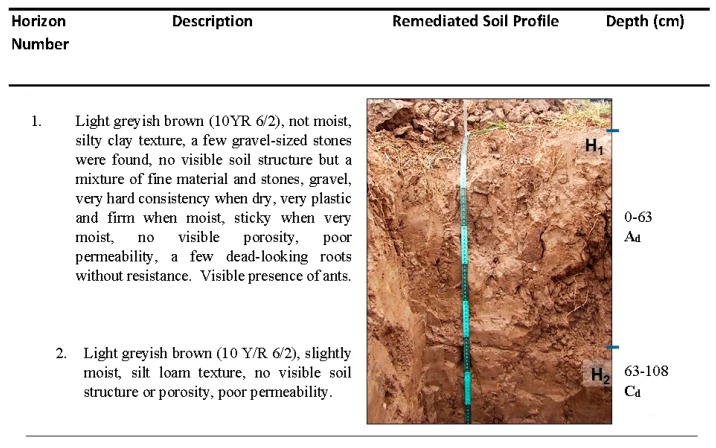
Remediated soil profile.

**Figure 3 ijerph-17-00382-f003:**
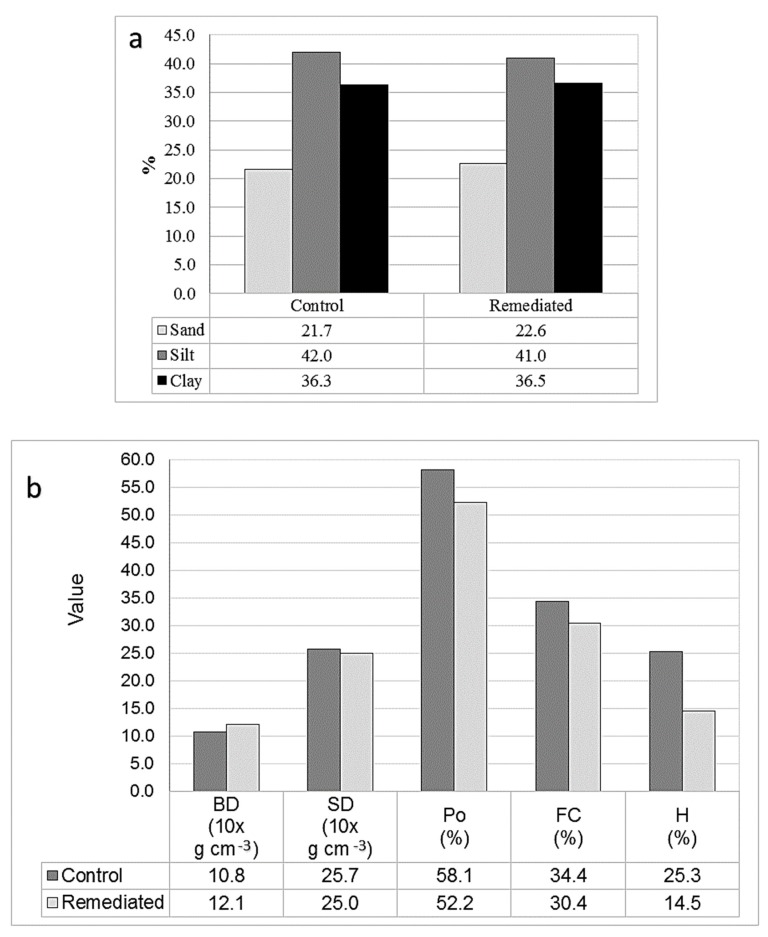
Control and remediated soil profile (weighted averages of 0–3 m soil depth) (**a**) texture and (**b**) bulk density (BD), particle density (SD), porosity (Po), field capacity (FC), and moisture content (H) values.

**Figure 4 ijerph-17-00382-f004:**
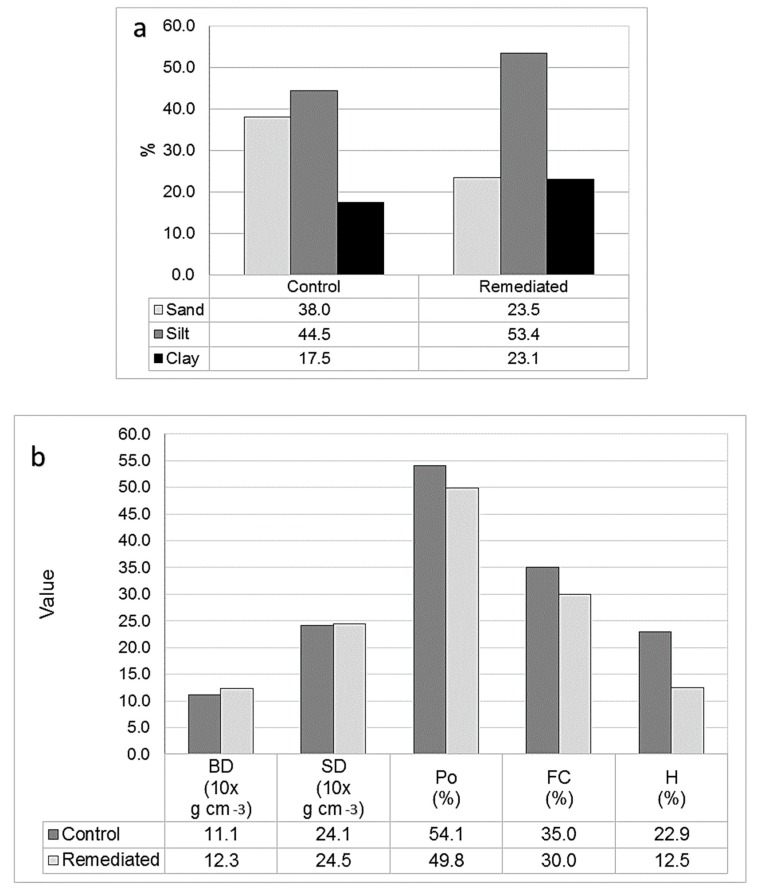
Control and remediated surface soil (0–30 cm, averages n = 5), (**a**) texture and (**b**) bulk density (BD), particle density (SD), porosity (Po), field capacity (FC), and moisture content (H) values.

**Figure 5 ijerph-17-00382-f005:**
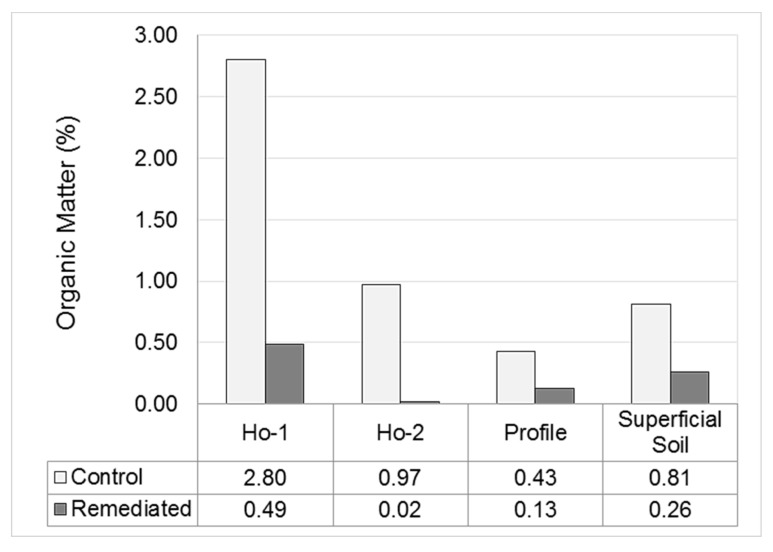
Control and remediated soil horizons, soil profile (0–3 m) and surface soil samples (0–30 cm), soil horizon 1 (Ho-1), soil horizon 2 (Ho-2), soil profile from 0–3 m depth (Profile), superficial soil for 0–30 cm depth (Superficial Soil).

**Figure 6 ijerph-17-00382-f006:**
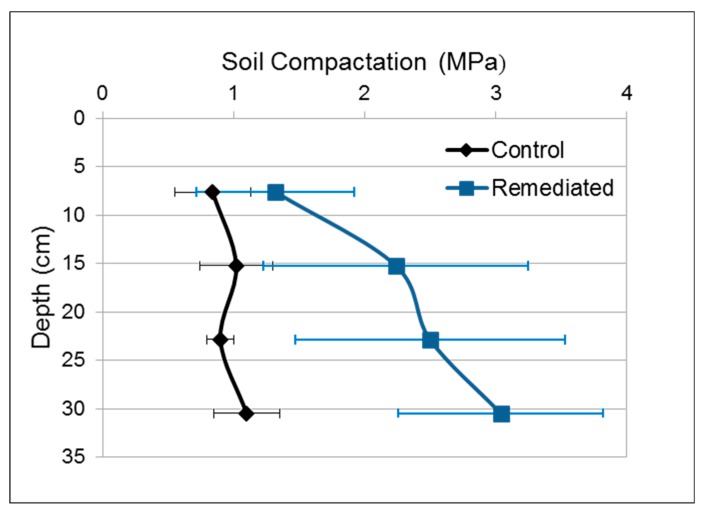
Control and remediated soil compaction, error bars represent one standard deviation.

**Figure 7 ijerph-17-00382-f007:**
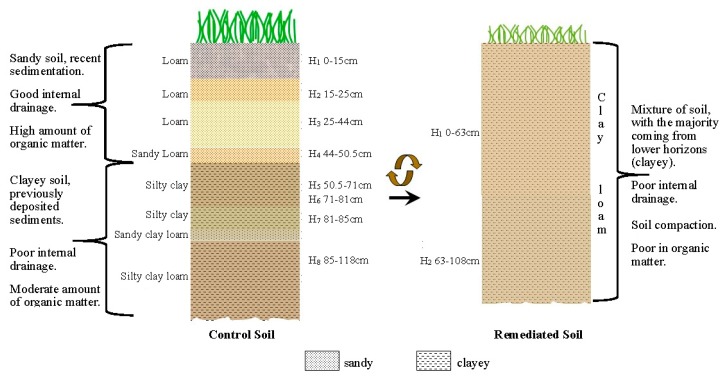
Pre- and post-remediated soil profile display.

**Table 1 ijerph-17-00382-t001:** Control soil horizons characteristics.

Ho	BD (g cm^−3^)	SD (g cm^−3^)	Po (%)	FC (%)	H (%)	Sand (%)	Clay (%)	Silt (%)	Texture (USDA)	OM (%)	pH	WDPT (s)
H_1_ (Ap)	1.06	2.38	55.31	35.88	24.69	50.46	9.34	40.20	Loam	2.80	7.08	0.59
H_2_ (A_1_)	1.16	2.50	53.76	32.44	16.31	42.96	17.41	39.63	Loam	0.97	7.98	0.48
H_3_ (C_1_)	1.20	2.63	54.40	31.23	11.17	49.04	17.34	33.62	Loam	0.16	8.01	0.52
H_4_ (C_2_)	1.28	2.63	51.51	28.30	18.87	67.38	13.12	19.50	Sandy loam	0.09	8.11	0.52
H_5_ (C_3b_)	1.05	2.50	58.08	34.19	26.08	9.60	47.41	42.99	Silty clay	0.16	8.14	2.36
H_6_ (C_4b_)	1.03	2.78	62.99	36.10	27.64	7.02	45.62	47.35	Silty clay	0.29	8.16	2.33
H_7_ (C_5b_)	1.22	2.38	48.59	31.51	24.33	57.08	15.64	27.28	Sandy clay loam	0.42	8.40	0.76
H_8_ (C_6b_)	1.04	2.63	60.63	35.97	29.06	15.08	37.64	47.28	Silty clay loam	0.43	7.91	2.27

Horizon (Ho), Bulk Density (BD), Particle Density (SD), Porosity (Po), Field Capacity (FC), Moisture Content (H), Organic Matter (OM), Water Drop Penetration Time (WDPT): Water, Edaphogenetic Processes: Pasture (p), Buried (b).

**Table 2 ijerph-17-00382-t002:** Remediated soil horizons characteristics.

Ho	BD (g cm^−3^)	SD (g cm^−3^)	Po (%)	FC (%)	H (%)	Sand (%)	Clay (%)	Silt (%)	Texture (USDA)	OM (%)	pH	WDPT (s)
H_1_ (A_d_)	1.25	2.50	49.92	30.93	10.05	30.36	32.64	37.00	Clay loam	0.49	7.50	1.26
H_2_ (A_d_)	1.18	2.50	52.80	30.23	15.77	20.36	37.64	42.00	Clay loam	0.02	7.80	1.97

Horizon (Ho), Bulk Density (BD), Particle Density (SD), Porosity (Po), Field Capacity (FC), Moisture Content (H), Organic Matter (OM), Water Drop Penetration Time (WDPT): Water, Edaphogenetic Process: Compaction (d).

**Table 3 ijerph-17-00382-t003:** Coefficient of determination (R^2^) between residual TPH concentration and soil physical chemical properties.

Parameters	Control Soil	Remediated Soil
BD (g cm^−3^)	0.8100	0.3721
SD (g cm^−3^)	0.7500	0.5329
H (%)	0.3400	0.3969
FC (%)	0.0049	0.2209
Po (%)	0.6400	0.1936
OM (%)	0.0900	0.0081
pH	0.0841	0.4489
WDPT (s)	0.1024	0.6889
Sand (%)	0.1521	0.0036
Silt (%)	0.1764	0.0001
Clay (%)	0.0001	0.0324

Bulk Density (BD), Particle Density (SD), Moisture Content (H), Field Capacity (FC), Porosity (Po), Organic Matter (OM), Water Drop Penetration Time (WDPT). These values are from surface samples (0–30 cm depth) with five replicates for each area (control and remediated).

**Table 4 ijerph-17-00382-t004:** Summary and Comparison of Fertility Parameters in Surface Soils (0–30 cm).

Parameters	Control Soil	Remediated Soil	Importance
Parameters very likely related to soil degradation from contamination or remediation technique			
Proportion fine to coarse particles (clay + silt):sand	62:38	77:23	May cause compaction, reduced infiltration, reduced root penetration, reduced gas exchange
OM (%)	0.81	0.26	Reduces CEC, CIC, may reduce moisture content and availability of soil nutrients
FC (%)	35.0	30.0	Reduces moisture retention, may cause water stress, wilting
Po (%)	54.1	49.8	May cause compaction, reduced infiltration, reduced root penetration, reduced gas exchange
BD (g cm^−3^)	11.1	12.3	May cause compaction, reduced infiltration, reduced root penetration, reduced gas exchange
Compaction (MPa)	0.84–1.10	1.32–3.04	Reduces water infiltration, root penetration, free gas exchange (respiration of soil organisms); about two to three times greater in remediated soil
H (%)	22.9	12.5	May cause water stress and wilting; about half as much moisture in remediated soil
Depth of roots (soil profile, cm)	118	3–10	Sign of unfertile conditions for plant growth, possible due to compaction
Presence of insects and spiders (soil profile, cm)	0–118	0–63	Sign of poor conditions, possibly due to poor plant growth (root penetration, primary productivity) and less food available
Parameters very likely not related to soil degradation from contamination or remediation technique			
TPH (mg kg^−1^)	182	969	Low levels, no significant correlation found between TPH and other factors (R^2^ < 0.7) in remediated soil
pH	7.1–8.0	7.4–8.5	Mildly alkaline but in same range as subsurface of control soil (7.9–8.4); probably not detrimental to soil fertility
WDPT (s)	0.48–0.59	1.26	Levels classified as “null”
Toxicity	NA	NA	No relationship was found between the soil concentration in the bioassay and response of the test organisms (all samples considered non-toxic)

Soil Organic Matter (OM), Field Capacity (FC), Porosity (Po), Bulk Density (BD), Moisture Content (H), Total Petroleum Hydrocarbons (TPH), Water Drop Penetration Time (WDPT), not applicable (NA). Note: depth of roots and presence of ants and spiders is presented for the entire soil profile, not just 0–30 cm.

**Table 5 ijerph-17-00382-t005:** Additional parameters to consider in future studies.

Parameters	Importance
CEC (meq kg^−1^)	Low levels reduce availability of soil nutrients, especially related to low OM
Salinity (dS/m)	Some remediation agents could increase salinity; probably not a factor in this study considering neutral—mildly alkaline conditions (high salinity is usually associated with high pH)
Microbial biomass/respiration (CFU g^−1^; mg CO_2_ h^−1^ kg^−1^)	Extreme pH, oxidizing conditions, and high surfactant concentrations may reduce microbial biomass, activity and important soil functions; this may not be a factor at this site considering the five year time span since remediation, humid tropical climate, and Fluvisol conditions (generally optimal for soil recovery)
Plant bioassay	This is a true confirmation of successful site remediation; a previous in situ study at this site with radish did not show reduced emergence, establishment or vigor, but bulb diameter was much less in the remediated soil; possibly due to soil compaction

Cation Exchange Capacity (CEC), Colony Forming Units (CFU).

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
