# Peer review of "Fertility Deterioration in a Remediated Petroleum-Contaminated Soil"

_ijerph, 2020, doi:10.3390/ijerph17020382_

Round 1

Reviewer 1 Report

English language and style should be revised in such a way that meaning of the sentences become clear to the readers. In that case very long sentences can be written in several separate sentences. example: line number 2, 26,35,47-50, 93-96, 164-166,229-231, Moreover, some sentences seem to be incomplete. Example: line No 20, 120, 127, 132-133,167,216. 

Some references are mentioned in the text in an inappropriate way which should be revised. Example: 74, 85, 92, 159.

Figure's numbers are not appeared in the text sequentially as per the journal's guideline. For example: Fig 3a then Fig. 5 leaving Fig. 4. Moreover, figure number a/b is not mentioned in figure 4. Figure 3b, 4b are not cited in the text. 

Sub-section 4.2 is not found the text. Conclusion part requires a very brief summery of the obtained results. 

The method of taking images in figure 1 and 2 should be clearly mentioned.

Elaboration of SOM (line:190) should be mentioned.

The cause of TPH concentration lowering of the sites under investigation is not explained (Line 256-257).

It is stated in line no 258 and 259 that "In many sites, residual TPH concentrations can be damaging in soil, impacting physical-chemical properties and resulting in a reduced soil fertility [3, 5]. However, this does not appear to be the case at this site." An explanation is required in supporting this statement. Standard fertility parameter should be mention in the text to compare the deviation of your result with the standard one (line no. 265).

"Remediated soil fertility damage was not due to residual hydrocarbons at this site, but instead to the mixing of lower horizons (silty clay to silty clay loam) with upper horizons (loam) during the remediation process" (line 309-310). An explanation is required in the result/discussion section stating the reasons for which the fertility of the remediated sites decreases.

Author Response

Response to Reviewer 1:

We would like to thank Reviewer 1 for his or her recommendations. We have attended to the long and incomplete sentences mentioned. Likewise, we have corrected the order in which the figures are presented in the text and the a/b in Fig 4. As suggested, section 4 was separated into 4.1 and 4.2. In the methods section, the methods used to make figures 1 and 2 are mentioned. The meaning of SOM is explained. Probable causes for TPH lowering has been presented. A more explicit explanation as to why the TPH concentrations are not damaging to soil (in this case) is presented. The fertility parameters in line 265 have been more clearly mentioned. The discussion was greatly amplified to present reasons for decreased fertility in remediated soils in a much clearer and explicit manner. Thank you.

Reviewer 2 Report

I carefully read through the authors' paper and found that the work on topic selection, questioning, analysis (methodology) and interpretation is correctly formulated and carefully presented - however, with a revision of the paper being recommended in view of the follow-up remarks.

In fact, it is important, inter alia to further investigate the sterilization of soils after their decontamination with oxidants, especially since this chemical method leads to partial or complete infertility (death of the microorganism populations), thus because of the side-products of the chemical reactions which causes high pH values ​​(not uncommonly 11 up to 13). It can take years to settle a normal pH in well-aerated soils, and much longer in clay-structured soils, allowing the soils to regain a normal population of microorganisms.

For this reason, it would be advisable to describe the used method of chemical oxidation in more detail. Usually, this method is namely carried out not only with hydrogen peroxide, but with an additional catalyst. Of interest would be the concentration of these used chemicals. Furthermore, the method of soil washing should be described.

Author Response

Response to Reviewer 2:

We have taken into consideration the observation by Reviewer 2 with respect to possible pH changes in the soil, especially in a greatly amplified discussion section. At this site, in the surface soil, the pH was only slightly alkaline and in the same range as subsurface soil from the control. This may be due to the long time for recovery (five years) and the moist, tropical monsoon climate at the site. With respect to knowledge about the remediation method, unfortunately this was not available. The company that did the remediation did not wish to divulge this, claiming that it was proprietary information (industrial secret). All we know is the little bit gained from one ex-employee from the company and the property owner’s observations. This is mentioned more explicitly in the methods section. We thank Reviewer 2 for his or her thoughtful observations.

Reviewer 3 Report

The topic is interesting but more data are necessary to confirm or not the most important hypoteses. I suggest to make some agronomic experiment for evaluating the effect of fertility losses of remediated soils on: plant growth, root length, nitrogen uptake, soil respiration, soil micorbia biodiversity,.....

Futhermore speficic experiments must be made to verify if some fertilization or soil tillage protocol could restore fertility of such soil.

Lines 284-285. “Vegetative growth was very probably limited also, not only by these factors, but by the lack of nutrients as well”., but in the results there are not data about vegetative growth.

This data are necessary  to confirm or not the hypothesis of this paper about the decrease in the fertility due to remediation techniques

Lines 286-287. “Although the use of organic fertilizers as post-remediation alleviators to the previous problems were not considered in this remediation project, it is very likely that they would have ameliorated negative soil impacts”.

Also this hypothesis must be conformed with specific experiments and data.

Line 304. “Soil conditioners and a mid-term vegetative cover (2-3 years) should be incorporated at the remediation site. “

Also this hypothesis must be conformed with specific experiments and data.

Lines 309-310. “Remediated soil fertility damage was not due to residual hydrocarbons at this site, but instead to the mixing of lower horizons (silty clay to silty clay loam) with upper horizons (loam) during the remediation process”.

Soil fertility decrease could be due to many other factors. In this paper there are no data that can allow to ascribe to only this factor the loss in fertility.

Other experiments with plants (e.g. analyzing growth, root length, nitrogen uptake, soil respiration, soil micorbia biodiversity,……) can help you to find the true causes of fertility losses.

Author Response

Response to Reviewer 3:

We agree that more data would be helpful to better understand what other factors may have caused the deterioration in soil fertility. These have been explored to a much greater extent in the discussion, which was amplified by several pages. Likewise, the explanations presented have been couched in wording like “probable”, “very likely” etc. without insisting that the reasons presented here are necessarily the only ones or predominant ones to explain the data. With respect to further experiments with plants, in the discussion section is it more explicitly mentioned that at this site there has already been such an in situ bioassay, with radish plants. Also, some indication with respect to biotic factors observed at the site, such as root penetration and presence on insects to depth are more explicitly mentioned in the discussion section. Considering the observations and recommendations on further experiments with organic conditioners, nutrients, and leaving the site under vegetative cover, the reasons for these recommendations were based on actual field experiments on similar sites in this tropical region of Mexico. These experiments have been more explicitly cited in the discussion section. We thank Reviewer 3 for his or her observations to improve the quality of this manuscript.

Reviewer 4 Report

The authors compared the remediated soil (soil washing and chemical oxidation) with the control soil to examine the effect of the remediation on the soil fertility. However, the parameters the authors examined are neither relevant nor enough to achieve the purpose. Most importantly, the soil fertility is not determined by the parameters the authors determined. The authors should be encouraged to conduct the additional study to describe the soil fertility more rigorously.

The parameters measured in this manuscript

Soil profile Physical properties of soil: bulk density, particle density, porosity, field capacity, moisture content, soil texture, water drop penetration time (soil water repellency), soil penetration resistance (compaction) Chemical properties of soil: organic matter content, pH

However, the soil fertility should be considered as (1) the potential to supply nutrients and (2) water retention/conductivity suitable for the agricultural practice. Therefore, the authors should add these parameters for the evaluation.

Parameters in relation with nutrient supply: N, P, K contents, electrical conductivity, cation exchange capacity, inorganic nitrogen production from soil organic matter, microbial biomass Water retention/conductivity suitable for the agricultural practice: hydraulic conductivity, three phase at field water capacity

In addition to the points listed above, the remediation technologies applied are soil washing and chemical oxidation. Therefore, it is natural that the fertility decreased in the remediated soil. The authors should describe what is new findings in the study.

That is all.

Author Response

Response to Reviewer 4:

Reviewer 4 mentions that soil fertility is determined basically by:

1) the ability of the soil to supply nutrients to plants and

2) to supply adequate water to the plants.

3) We would add that also, that the soil should not be toxic.

We are in agreement with this reviewer on these points. We believe that the best way to determine if the soil can supply nutrients to plants is the Cation Exchange Capacity (CEC), and also, compaction (which affects root penetration).

The amounts of N-P-K can be quite variable, according to fertilization plans, recent rainy or dry weather, cropping, etc. But the ability to maintain nutrients in the soil over the long run depends on CEC. This in turn is principally driven by the kinds and amounts of clays and organic matter in the soil, and to some extent, the health of the microflora to transform and supply nutrients from the soil in forms that are utilizable by plants.

Taking this into account, in a greatly expanded discussion section, these considerations are explained much more explicitly. Based on studies of alluvial soils in the region, it is explained that they are generally high in CEC, and that the increase in clay in surface soil in the remediated site would also increase the CEC – thus, it is unlikely that CEC could be less in the remediated soil than in the control.

Likewise, the importance of compaction and its general relationship to root penetration (and the ability of roots to obtain nutrients and water) is more explicitly stated in the discussion. In the remediated soil, the compaction was two to three times greater than in the control soil, and root penetration was roughly 10-15% of that in the control soil, according to observations in soil profiles.

In this same context, the non-toxicity of the soil (which could affect microbial growth and nutrient availability) is discussed, as well as some ways to measure microbial numbers of activity in future studies. Also in this grain, the site history in terms of the generally fertile properties of alluvial soils in the region, the moist, tropical monsoon climate, and the five year time lapse between the commercial remediation and sampling (really optimal conditions for recovery) have been presented as factors favoring recovery.

Additionally, with respect to water availability, we mention explicitly in the discussion section that in this soil there was null water repellency, although, due to compaction, it is much less probable to obtain good infiltration. Likewise, in the field, the actual moisture content was only one-half that of the control soil.  

Thus, with different parameters than those mentioned by Reviewer 4, and citing references of studies on regional soils and recovery projects (including an in situ radish plant bioassay at this very site), we have addressed these issues, and have greatly expounded on them in a much more explicit way in the amplified discussion section.

We believe that even though surfactants and chemical oxidizers may damage the soil biota (in the short term), if the soil is truly health (in chemical and physical terms), this should be overcome. This would be especially so considering the five year time elapsed between remediation and soil sampling, considering the high CEC of the alluvial soils in this region, and the excellent, tropical monsoon climate which greatly favors biological activity. These arguments are expanded on and made much more explicit in the amplified discussion section of the revised manuscript.

What we find truly novel in this study is the focus on soil fertility as a remediation goal, how to measure it, using those methods in the study and other proposed methods, the lack of toxicity or water repellency in the soil, and the non-correlation between residual TPH and fertility parameters. Also, the focus on additional methods during and post- remediation to assure that these adverse effects are avoided or minimized, is stressed.

We greatly thank Reviewer 4 for his or her criticisms of this manuscript. These have enabled us to address these uncertainties and explain them in a much more concise and explicit manner. We consider that this has really improved the quality of the discussion and the overall manuscript. Again, thank you.

Round 2

Reviewer 3 Report

The added information and data allowed to improve the significance of this interesting paper.

Nevertheless, some further corrections are still necessary in table 4.

Bulk density (g cm-3) values must be corrected in 1.11 and 1.23.(or if you prefere change the unit in 10 x g m-3)

Some details of methods used for measuring depth of roots and presence of insects must be added in materials and methods sections.

Author Response

We are in agreement with these corrections, which have been made in the text as per Reviewer 3. The change in units was made in both figure 4 and figure 3 for consistency. We used 10x g cm-3 to be able to have values of approx. 10-20 and to present them in the same figure with the other values (presented in percent). If we had used just "g cm-3" the values would have been too small to visualize well in the same figure. We have added a reference to support the methods section with respect to the soil profile and made a brief description of this method in the text (Cuanalo, 1990, ref. no. 14). Changes are highlighted in yellow in the revision copy.

We thank Reviewer 3 for his or her careful attention to detail to improve the quality of this manuscript.

Reviewer 4 Report

The authors describes one case to consider the soil fertility in the remediation of soil contaminated with petroleum hydrocarbons and the recommendations. The revision made the manuscript understandable, in which the soil compaction is considered as an important parameter to show the deterioration of soil fertility in the case of this soil. However, it is usual that the compaction happens to some extent in the soil excavation, treatment and refill at the site. The authors should describe clearly what new findings are. One of the examples would be the mechanism of promoted compaction by mixing upper and lower horizons. There are also many parts that should be revised. I cannot point out them all because they are too many. The followings are some of them as examples.

Section 4.2 “recommendations” is described to be applied to the soil remediation generally. However, the conclusions obtained here is only applicable to the case the authors describe. For the “recommendations” many cases of soil remediation are needed to be studied. This section should be deleted. When the soil excavation and chemical oxidation is carried out to decompose the petroleum hydrocarbons in the contaminated soil, the deterioration of soil fertility would not be considered as the authors pointed out. Therefore, the soil restoration treatment to increase soil fertility is usually carried out after the remediation of farmland. On the other hand, in the case that the contaminated soil is used for constructing the building, the compaction of soil might be considered as beneficial effect. The reviewer reads the manuscript with assumption that the contaminated field is farmland, but the manuscript does not show it clearly. The authors should describe the information on the land use, time after the remediation, climate at the site, and any treatments after the remediation for soil fertility. The restoration of soil fertility in the compacted surface soil is well known as the authors pointed out in Conclusion: the incorporation of organic matter (farmyard manure and etc). When the heavy machine is used to excavate the soil, the compaction of soil to some extent cannot be avoidable. Therefore, what the authors conclude “do not mix the lower and upper horizons” does not make sense. The relation of the “mixing” and “compaction” should be described for this conclusion. Although the compaction of soil is shown as the important parameter to indicate the fertility of the soil, the authors do not show how the compaction has happened in the remediated soil but just speculated. Again, what the authors conclude “do not mix the lower and upper horizons” does not make sense. The relation of the “mixing” and “compaction” should be described for this conclusion. Figure 3 shows the results both as figures and tables. The same data are repeated. Please select one to show. There is no statistical information in these values. The authors should be encouraged to show them if they are significantly different between control and remediated soils. The soil samples used for the analyses are not clearly shown (It would not be the soil samples obtained from individual horizons.), which should be described in the caption. Comments on Figure 4 are just the same as ones for Figure 3. Figure 5 shows the comparison of the organic matter content between control and remediated soils. However, there is no description for the abbreviation, Ho-1, Ho-2, Profile and Superficial soil. These abbreviations should be described in the caption. The table and figure show the same values repeated. Please select one of the two. Please add the statistical analysis (standard deviation/error and significant difference). Figure 6 shows the soil compaction with soil depth. However, the measured point at depth 15cm in control soil is located on the interface of H1 and H2 horizons. It does not make sense. The standard deviation of the soil compaction analysis should be also shown. Table 3 shows the regression coefficient between the residual TPH concentration and the soil physical chemical properties. However, the residual TPH concentration is only two concentration in control and remediated soils. The correlation analysis with only two data does not make sense. The authors should clarify how the correlation analysis was carried out. In the case there are more data on the residual TPH concentration, those data should be shown. The soil samples used for this analysis should be also shown in the caption or footnote. Table 4 shows the values of various parameters. Some parameters are shown with only one figure without standard deviation/error and others are with the range of values. The authors should show one figure with standard deviation/error. In the case of range, the authors should add the explanation what it means. In the case of depth of roots, the authors should describe what “118+” means.

That is all.

Author Response

Please find response in uploaded document.

Thank you.
